# Evaluating Consumer Behavior, Decision Making, Risks, and Challenges for Buying an IoT Product

**Majid Nasirinejad** and **Srinivas Sampalli** *

Faculty of Computer Science, Dalhousie University, Halifax, NS B3H 4R2, Canada
* Correspondence: srini@cs.dal.ca

**Abstract:** Home appliance manufacturers have been adding Wi-Fi modules and sensors to devices to make them 'smart' since the early 2010s. However, consumers are still largely unaware of what kind of sensors are used in these devices. In fact, they usually do not even realize that smart devices require an interaction of hardware and software since the smart device software is not immediately apparent. In this paper, we explore how providing additional information on these misunderstood smart device features (such as lists of sensors, software updates, and warranties) can influence consumers' purchase decisions. We analyze how additional information on software update warranty (SUW) and the type of sensors in smart devices (which draw attention to potential financial and privacy risks) mediates consumer purchase behavior. We also examine how other moderators, such as brand trust and product price, affect consumers' purchase decisions when considering which smart product option to buy. In the first qualitative user study, we conducted interviews with 20 study participants, and the results show that providing additional information about software updates and lists of sensors had a significant impact on consumer purchase preference. In our second quantitative study, we surveyed 323 participants to determine consumers' willingness to pay for a SUW. From this, we saw that users were more willing to pay for Lifetime SUW on a smart TV than to pay for a 5-year SUW. These results provide important information to smart device manufacturers and designers on elements that improve trust in their brand, thus increasing the likelihood that consumers will purchase their smart devices. Furthermore, addressing the general consumer smart device knowledge gap by providing this relevant information could lead to a significant increase in consumer adoption of smart products overall, which would benefit the industry as a whole.

**Keywords:** Internet-of-Things; purchase behavior; smart devices; privacy

## 1. Introduction

People are purchasing an increasingly wide variety of smart products—from voice assistant devices (e.g., Google Nest and Alexa Echo) to security systems (e.g., smart doorbells and smart locks) and from small kitchen appliances to major home appliances in the early 2020s. It is estimated that the market demand for "home automation"—smart products in the home that are monitorable and controllable over the Internet (a.k.a Internet of Things (IoT))—increases by 21% each year [1]. Furthermore, the same pattern is forecasted for the future—making it one of the most widespread and profitable home product categories on the market [1].

Consumer IoT smart products are designed to replace our existing 'unconnected' products and appliances—saving time by remotely automating traditional manually performed tasks. These smart devices are becoming more affordable and accessible—as evidenced by the price of a smart light bulb dropping from USD 60 in 2012 to just USD 5 in 2021. As a result, millions of people can now purchase a wide range of smart products at low costs.

Experts forecast that 8.4 billion voice assistant devices (VA) will be used by 2024 [2]. These devices use a built-in microphone and speaker to perform tasks, including playing

music, controlling other smart home devices, and even online purchasing [3–6]. Approximately 147 million units of VAs were sold in 2019—70% more than in 2018. Interestingly, however, the number of Amazon Echo and Google Home (nest speaker) devices sold both reduced by 7% over the same time period [1]. This is likely because other smart home devices already had built-in voice command systems. Indeed, some of them, such as TVs and streaming modules, now have these voice assistant platforms fully integrated and accessible via the device's remote control [7].

Regardless of communication protocols and the connected network, IoT devices are made of two parts, hardware and software. Firmware is a program that is preloaded to the memory of the hardware of IoT devices during the manufacturing process [8]. Whether it is a smart light, smart TV, or smart coffee machine, all IoT devices use firmware for responding to or controlling smart device sensors, actuators, mechanical parts, and electronic components. There are multiple ways to interact with and control IoT devices, including using a mobile application [9], desktop application [10], voice control input [11], gestures [12], or web-based applications [13]. However, for some of these, the device needs to use specific sensors. For instance, voice command input requires a microphone, or a gesture controller needs a proximity or camera sensor.

In the last decade, research has been done to evaluate, report and review IoT devices from security perspectives by focusing on the device itself, the connected network, and communication protocols—mostly related to cyber-attacks such as Distributed Denial of Service (DDoS) that can be done through IoT devices. For instance, Google Scholar returns more than 525 published papers for "IoT DDoS" and 2600 results for "IoT privacy" from 2018 to 2022. However, the intersection between IoT devices from the user purchase behavior and technical perspective is not well-researched to date.

In 2022/2023, smart technology is still relatively new compared to traditional unconnected or mechanical products, and it is expected that customers are not completely aware of the full functionality of these smart products. Yet, the extent of consumer knowledge about smart devices is still limited [14,15]. Addressing this knowledge gap could lead to a significant increase in consumer adoption of smart products and a huge increase in market potential for both existing and new IoT products.

Our research explores how improving consumer understanding by providing additional information about smart devices will affect purchase behavior. Specifically, the purpose of this study is to explore the IoT purchasers' conceptions, misconceptions, and concerns about privacy, performance, and financial risks and the steps customers take to address these concerns.

Research shows that [16] the consumer decision-making process involves five steps: (1) need recognition, (2) information search, (3) evaluations of alternatives, (4) purchase behavior, and (5) post-purchase behavior. Our research focuses on steps 2 to 4. As such, our findings can serve as a guide for product designers, developers, and marketers alike. This enhanced understanding of customers' needs will facilitate more effective customer communication, making it more likely that they will purchase and enjoy new smart products.

Customers generally do not have access to risk information such as privacy, financial, and performance risks when purchasing IoT devices. Emami-Naeini et al. [17] found that IoT buyers do not consider privacy and security prior to purchase. Our research explores how customers' lack of knowledge about software updates and sensors operation impacts their purchase behavior.

The paper focuses on the following research questions:

- What is the impact of adding a software update warranty on the package of smart home appliances on consumer behavior?
- Does a sensor information label influence consumer behavior in purchasing a smart device?

From this, we can provide information to IoT manufacturers to help them better understand what information customers need and what will help them be seen as a trusted brand.

We organize the remainder of this paper as follows. Section 2 discusses the current literature on the IoT, consumer behavior, and decision making. Section 3 outlines the methodology that we used in our study and the experiment design. Section 4 explains the experimental tasks and results in detail. We discuss results and future research ideas in Section 5. Section 6 concludes the paper.

## 2. Literature Review

Household appliance industry leaders manufacture smart appliances, and market revenue is expected to be USD 71.1 billion in 2025—an increase of 244% from 2020 [18]. Clearly, there is a huge demand for smart appliances. However, simply adding a Wi-Fi module to a 'non-smart' device is not enough for consumers to perceive it as a desirable smart product.

### 2.1. Internet-of-Things (IoT)

Mark Weiser [19] states that ubiquitous computing (ubicomp) devices, such as tabs, pads, and boards or other IoT devices, require three features to be desirable to consumers: be cheap, have low power consumption, and an embedded software system. He states that ubicomp devices need to make everything work "faster and easier".

IoT product functions involve the following: an internet connection (direct or via a hub); device-to-device data transfer; storage of generated and collected data locally (or in the cloud) for further analysis; and remote control and monitoring of mechanical and electrical devices.

From a basic device (e.g., a smart light) to high-tech products (e.g., a Tesla car or a smart home appliance), all IoT devices have embedded software and firmware, which, from the consumer perspective, is the invisible part of the product.

The opportunity to hold and touch physical goods strengthens a feeling of ownership [20–22], increases the product's valuation [23], and establishes confidence in the product's quality [24]. However, there are no tactile ways for consumers to get the same positive feelings for, and confidence in, the software of IoT devices. Moreover, users' general lack of knowledge about smart product software and firmware update requirements has caused some consumers to lose confidence in certain smart product manufacturers after purchasing, as illustrated by this next example.

When Toshiba stopped its partnership with Google, its smart TVs with built-in Chromecast did not receive the Chromecast firmware updates the second year after purchase. This is to be expected for IoT products, for which manufacturers continuously try to add new smart features to their products without considering the impact on the lifecycle of the product. However, as previously discussed, consumers are generally unaware of these kinds of software changes that decrease the lifecycle of smart home products.

Thankfully, some companies provide longer software support for their products. For instance, Tesla guarantees the permanent functionality of "autopilot" software through the entire life of their car—as long as the system was added at the time of purchase. In fact, some automatic updates actually add functionality to certain smart products. For example, in February 2022, Whirlpool updated the firmware of their smart ovens over-the-air to add air frying (a popular home appliance trend) to the menu of the device [25], the idea being that now users no longer need to purchase a separate air fryer as they now have that functionality though their smart oven software update. This is clearly a value add to the consumer.

However, not all over-the-air updates improve user experience. Some of them, such as iPhone updates, actually slow down performance on older devices, clearly decreasing functionality [26].

### 2.2. Consumer Behavior and Decision Making

Consumer behavior and privacy concerns have been explored to a certain extent in the literature—from home users [17,27–31] to travelers [32,33] and in IoT design frame-

works [34–37]. But other aspects of IoT devices, such as financial risks and performance risks on home users' adoption of IoT devices, have not been reviewed.

Previous research shows that consumers are not looking for all available information and that information may have a negative impact on their decision making [38,39]. However, it is important to understand which information needs to be presented to customers. Smart home buyers tend to see the potential benefits rather than the risks of smart devices [40]. They value saving energy, time, and money as well as having to do fewer tasks [41]. Hsu et al. [42] found that compared to privacy and cost, usefulness and enjoyment have a significant effect on determining IoT adoption. Furthermore, consumers also believe that software will not enhance product quality in IoT devices [43].

Wang et al. [40] found that financial costs are seen as less important factors in consumer adoption of IoT devices. However, in our research, we specifically focused on and explored how additional information such as software update warranty and list of device's sensors could affect the customer's purchase preference, and we have found that the type of IoT device and its use can impact the effect of cost on consumer's decision to purchase. We also found that the cost of the IoT device and the software update warranty also has a significant effect on IoT adoption. We explore these findings further in our results and summary sections.

## 3. User Study 1

### 3.1. Methodology

For this study, we recruited 20 participants to complete four tasks. Participants' ages ranged from 18 to 68 years *(M = 32, SD = 10.5)*, with 90% identified as male, and 75% reported they purchase the smart devices by themselves. All participants had experience buying IoT devices and had used them for at least 3 months. Our recruitment notice was sent to the university student mailing lists and shared on social media platforms (Facebook, Twitter, and LinkedIn). We also shared the notice on Facebook Marketplace and Kijiji in the categories related to smart home devices where the visitors are looking to buy, sell, or trade smart devices.

The study was conducted virtually using video conferencing and took about 90 min to complete for each participant. We asked participants to confirm their willingness to participate in this study. Prior to our one-on-one sessions, the participants completed a background questionnaire on a Google form to help engage them, gauge their background, and build the requirements for the in-task conversations.

We started the one-on-one sessions with pre-task questions prior to asking participants to perform the main study tasks. We then asked participants to look at pictures of a smart device and let us know how they decided to purchase that device. We then gave participants additional information about the smart device and asked whether or not their purchase decision had changed and why. Finally, we asked participants to complete a post-task questionnaire and rate their level of knowledge before and after this user study. During the study, we also asked open-ended questions based on the participants' responses or task completion to gain further insights into the reasoning behind their responses. Participants were compensated with a payment of USD 20.

We attempted to control for potentially confounding variables to mitigate any undue influences on the results of our study. For this reason, we added inclusion and exclusion criteria to the recruitment notice. For example, the participants must have experienced using at least one smart device for 3 months—excluding smartphones and activity trackers, as these are personal smart devices, and our study focus was on smart home devices. We clarified the terms we used in the study with our participants so they were fully aware prior to engaging—for example the meaning of IoT, and what a software update warranty is.

In this experiment, we compared consumer responses through a 2 (information: base, added info) by 3 (devices: robot vacuum, TV, fridge) within-subject user design. We predicted that consumers' product preferences would drop with extra information about the list of sensors on the device (privacy risks) and software update warranty (performance

and financial risks). User purchase preference was the core dependent variable, and additional information (warranty information, sensor information) was the independent variable. Financial risks, performance risks, and privacy risks were tested as simultaneous mediators of the manipulations' effects on user preference (Figure 1).

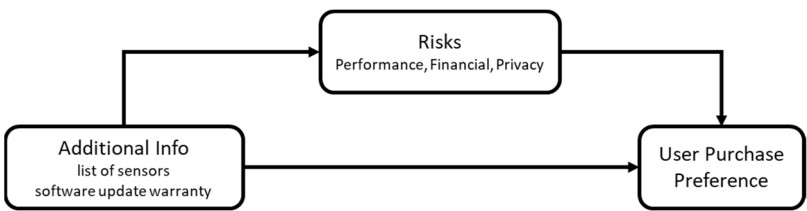

**Figure 1.** Overview of conceptual framework and user study.

*3.2. Tasks*

We selected three different smart home appliances (robot vacuum, TV, fridge)—each of which has its own properties, which are listed in Table 1.

**Table 1.** List of devices used in the task design with their properties.

| Device | Price | Branded | Easy to Replace? (Portable) | Sensors | Life Span | Others |
|--------|-------|---------|------------------------------|---------|-----------|--------|
| **Vacuum** | middle | no, similar | yes | camera, microphone, proximity | Mid | New gadget |
| **TV** | high | yes | no, bulky | microphone | Mid | High-end |
| **Fridge** | very high | yes | no, stationery | camera, microphone, proximity | Long | New features |

3.2.1. Robot Vacuum

Robot vacuums are portable, and they move independently around the house, cleaning floors. They are loaded with built-in tools such as: microphones and speakers for user interaction; proximity sensors such as IR sensors; digital 2D camera (with the ability to live stream) [44]; and laser technologies (for better navigation).

We selected a robot vacuum because they are a relatively affordable smart home device and quite popular with consumers [45]. Similar portable devices with heavily loaded sensors increases privacy risks in the home. We wanted to understand if buyers were aware of these risks at the time of purchasing the product. Figure 2 shows the pictures for this task which were shared with participants.

The following steps were considered in the interview:

- First, we asked participants if they had any experience using robot vacuums or if they had considered buying one.
- Next, we asked them to imagine that they were interested in buying a robot vacuum and that they found this item available at $399, which was within their budget.
- Then we displayed the front side of a robot vacuum package (Figure 2-top-left). The front side includes the name of the device and a picture of the device with a mobile phone, which indicates that the device can be controlled over the phone or via the Internet.
- Next, we displayed the back side of the package (Figure 2-top-center), which includes some specifications of the device, such as dimensions, weight, and battery type, but does not contain any information about the vacuum sensors.
- We asked participants to ask any questions they had about this product at this point.
- In the next step, we displayed the third image (Figure 2-top-right) and told them that a third-party company reviewed this device and added a sensor table to the package (Figure 2-bottom).

- In an open-ended form question, we asked them to let us know what they think about the sensor table and if the new information changed their decision to purchase this product.

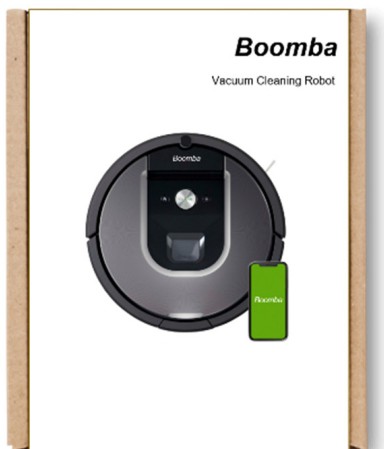 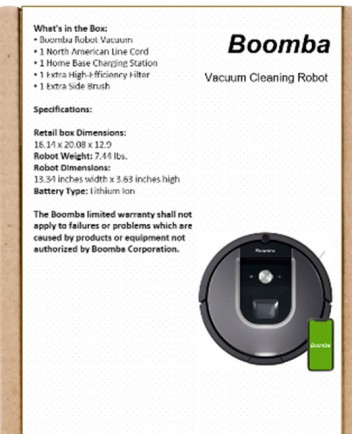 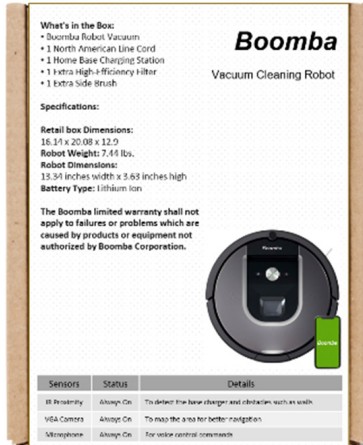

| Sensors | Status | Details |
|---|---|---|
| IR Proximity | Always On | To detect the base charger and obstacles such as walls |
| VGA Camera | Always On | To map the area for better navigation |
| Microphone | Always On | For voice control commands |

**Figure 2.** Front-side of the robot vacuum package (**top-left**) Back-side of the robot vacuum package without sensors table (**top-center**), with sensors table (**top-right**), sensors table on the package (**bottom**).

### 3.2.2. Smart TV

US households have at least one TV, and 70% of them are smart TVs [46]. We selected an expensive large (75″) branded (Samsung) smart TV. Many smart TVs have a built-in microphone, or they have a microphone on the remote control for user interaction with voice commands.

Smart TVs run firmware as an integrated OS, which is usually updated over the air. Most smart TVs are preloaded with popular streaming applications such as Netflix and YouTube. In addition, users can install other applications from the TV's app store. Application developers update their apps frequently, and in most cases, the content is no longer accessible if the user does not update the installed applications themselves.

For the same reason, the firmware of the TV needs to be updated to the latest edition. With the Microsoft Windows platform, customers delay software updates—mainly because of previous negative experiences [47]. However, on IoT platforms, most manufacturers remotely update users' devices over the air to the latest firmware every couple of months. Almost all of these new updates require more resources, from processor to memory, and this will cause the device to run more slowly after a couple of years. Eventually, that TV will no longer be able to receive the latest firmware update, and users will not be able to install the latest applications. This means that, after a few years, smart TV owners (who pay for both hardware (TV) and smart features (software)) will no longer be able to access the original smart TV features as the original hardware will not be able to cope with the updated software's added resource demands. The steps below are involved in the survey:

- In this task, we asked participants if they have a TV or not, the size of the TV, whether it is a smart TV or not, why they replaced their previous TV, and how often they replace their TV.

- Next, we displayed a digital ad for a TV (Figure 3-left) and asked them to imagine that they were interested in buying this smart TV and that the price was within their budget. The ad contains the size, quality of picture, brand, price, and manufacturer warranty of the TV.
- Then, we asked participants to ask any questions they had about this product.
- When they had no more questions, we told them that we had received a second ad with updated info from the manufacturer. Then, we displayed the second image (Figure 3-right), which contains everything in the first image plus information about the software update warranty, and asked them to let us know if they had any questions.

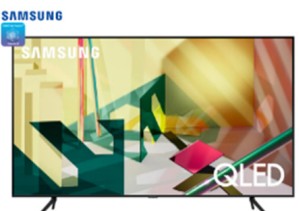
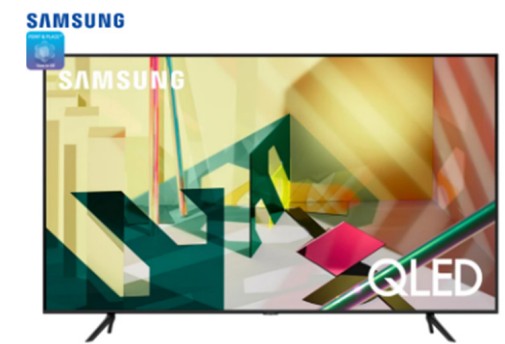

**Figure 3.** Smart TV ad, (**left**) without software update warranty, (**right**) same ad with software update warranty.

This task was designed to understand if consumers were aware of the software updates and software warranty. Furthermore, to ascertain whether they understood what they were paying for when purchasing a smart TV (or smart devices in general), specifically, that they are not paying only for the hardware and the device itself but paying for the software as well. We asked participants about their alternative solutions if the smart features of their TV were no longer working. We will explore their perspectives in the discussion section.

### 3.2.3. Smart Fridge

Major appliances such as refrigerators, ranges, and dishwashers are more expensive and are expected to work longer than small appliances such as a vacuum or coffee maker. Almost all major appliances are stationary, and customers generally do not relocate them due to the design of the cabinets, power supply location, and availability of water lines.

For this task, we selected a smart fridge with a 21.5" vertical screen. Compared to a non-smart fridge of the same size and capacity, it is about two times more expensive. The fridge has a built-in microphone for voice commands, an indoor camera to display inside the fridge, and other smart features through the screen. This part of the survey has the following steps:

- Before starting the task, we asked participants if they had any experience using a smart fridge or if they had ever considered buying one.
- We also asked them about the expected lifespan of the fridge and how often they are replacing their fridge.
- We started this task by displaying the first image (Figure 4-top-left) and asked them to assume that they were interested in buying a smart fridge and that they found this item available in a store that was within their budget. We asked them to bring up any questions they had.
- After a couple of questions and answers, we told them that we found a new brochure for this device, which was released yesterday, and the company added some more information. Then we displayed the second image (Figure 4-top-right), which included a sensor table and software update warranty.
- We asked participants to let us know what they thought about the new information provided. We wanted to understand what they thought about the longer software

update warranty. Moreover, we wanted to know if consumers were aware of how much they were paying for smart features, the smart features' performance, and the duration of the software update warranty. Additionally, we wanted to ascertain whether or not the added info would impact their decision to purchase.

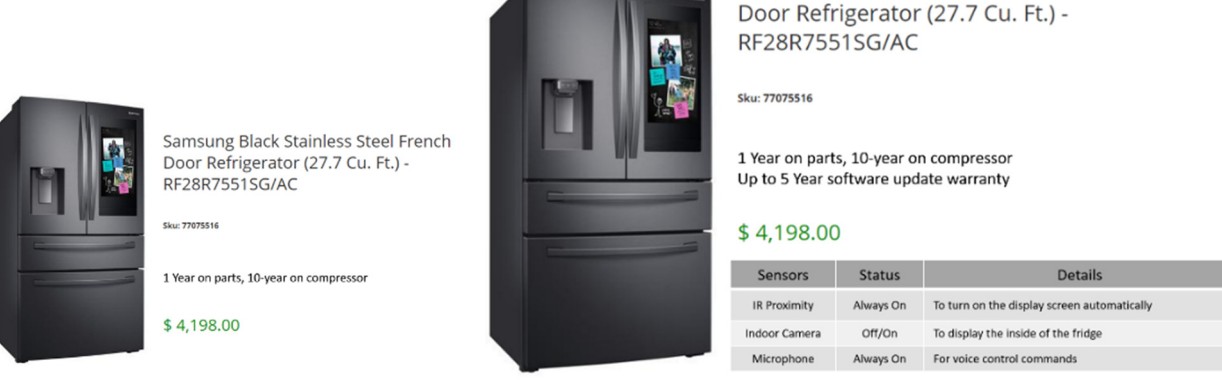

**Figure 4.** Smart refrigerator brochure, (**top-left**) without sensor table and software update warranty, (**top-right**) same brochure with sensor table and software update warranty, (**bottom**) sensors table.

### 3.2.4. Smart Light Bulbs

Once participants finished the main tasks, we shared a picture of two smart light bulbs side-by-side (Figure 5). We told them that these two bulbs have similar technical specifications and warranties. We asked them which one they would choose if they wanted to add 10 smart lights to their properties.

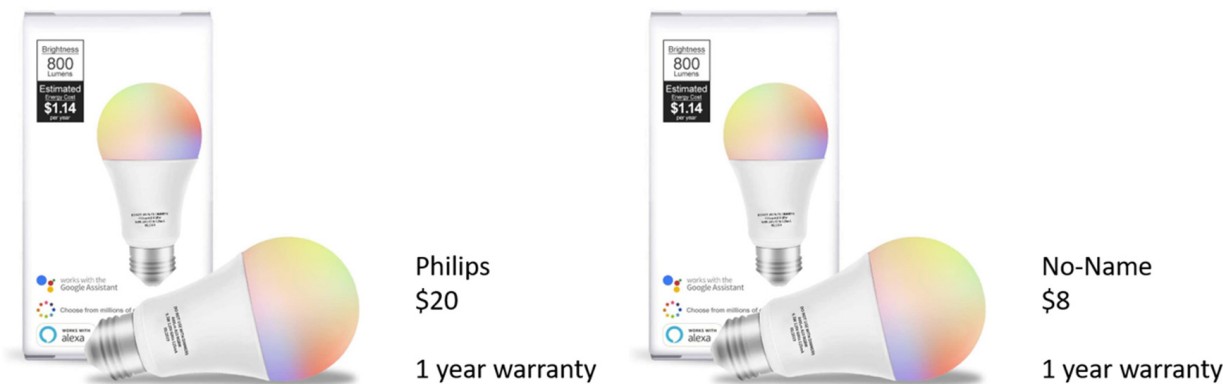

**Figure 5.** Smart Light Bulbs, (**left**) branded, (**right**) no-name brand.

The left bulb is a branded (Philips) light (a reputable name in the smart light market) and is priced at USD 20. Conversely, the bulb on the right is a no-name brand priced at USD 8 (2.5 times cheaper than the branded bulb). We asked them which lightbulb they would buy and why.

*3.3. Results*

To complete this section, we analyzed the interviews of the 20 participants who had owned an IoT device for at least three months. They introduced their devices (Table 2) to the researchers and answered open-ended questions about how they use them and how they decided to purchase the devices. This mix of quantitative and qualitative methods allowed us to understand consumer IoT usage from a richer range of perspectives and to better understand how and why people buying IoT devices make their purchase decisions.

**Table 2.** List of participants' smart devices.

| Device | Qty | Device | Qty | Device | Qty |
|---|---|---|---|---|---|
| Google home/Alexa Echo | 11 | Thermostat | 5 | Security System | 3 |
| Smart TV | 9 | Light Switch | 2 | Vacuum | 1 |
| Light Bulb | 7 | Chromecast/Fire TV | 3 | Tile tracker | 1 |
| Major Home Appliances | 6 | Garage Door Opener | 3 | Others | 2 |
| Plug | 5 | Doorbell/Door Lock | 3 | | |

We contribute a comparison of users' privacy perceptions before and after the study tasks. Specifically, we assess any changes in willingness to purchase smart devices once they are aware of the sensors that track and monitor with mics or cameras. In essence, we determine if this additional sensor information raises concerns for devices in relation to privacy risk assessment.

We contribute a comparison of users' financial perceptions in the form of willingness to know the device's life span and concerns about software updates in relation to the device's financial risk and the device category.

We compare the recognized functionality, performance, and device assessments of the two versions of devices (before and after software update) and provide qualitative insights explaining how to design better devices and features.

### 3.3.1. Robot Vacuum

While all participants were interested in purchasing the robot vacuum in the base condition (no additional info about the sensors), 10 participants changed their decision to not purchase the device when they found sensor data information (added info condition) on the new box of the product. Of the participants, 65%, including all 10 participants who changed their decision, were worried about the camera on this device as it is moving around the house. Of the 10 participants who changed their decision, seven had concerns about both the camera and the microphone. To illustrate, P13 said they scheduled the cleaning when the kids were not home because he had concerns about the kids' privacy.

### 3.3.2. Smart TV

A total of 35% of the participants had no idea what a software update meant, 90% of the participants did not know that the smart TV might not automatically receive the latest software update, and they assumed that their TV received software updates forever.

In a base condition (no additional info), all participants were interested in purchasing the smart TV. However, after reviewing the updated brochure with added information about the software update warranty, 50% of participants decided not to purchase that smart TV because of the only one-year software update warranty. While eight participants reported that they are expecting to get at least a 5-year software update warranty, seven of the participants believe that 2–3 years of software update warranty be fair enough. One participant believed that the warranty is not necessary and the device should receive the operating system updates for the life of the TV.

### 3.3.3. Smart Fridge

Participants believe that a fridge's lifespan is around 10 years ($M = 10.45$, $SD = 5.4$). In the base condition (no additional info), all participants were interested in purchasing the smart fridge, and all of them found that a 1-year warranty and a 10-year compressor warranty are standard for all refrigerators. In the added info condition, 25% of the participants changed their decision to not purchase this smart fridge because of the 5-year software update warranty. While no one had a concern about the indoor camera, five participants pointed out that they preferred to be able to control the microphone manually. However, other participants (P6 and P9) noted that the open mic is required for smart home devices to send commands and ask questions remotely, specifically for persons with disabilities. P17 was the only participant who had purchased a smart fridge, and until participating in this task, did not know that the fridge's software needs to be updated frequently.

### 3.3.4. Smart Light Tasks

Of the participants, 25% selected the branded light, and all of them said that they trusted the brand. The 75% of participants who selected the no-name brand gave two main reasons for their choice. The first is that a basic product, such as a smart light, has both similar performance and quality and offers the same warranty period. The second reason was that they felt the extra cost for the branded light was simply because of the brand itself and perhaps increased customer support. However, because these basic products do not generally require additional support, the extra price was not warranted.

P11 mentioned that if the no-name product had no warranty, he would purchase the branded one. However, as both products have a 1-year warranty, he would be able to replace the no-name device two times if it failed, and it would still be cheaper than purchasing one of the branded smart lights. P13 had a previous good experience with non-branded IoT products, so he was not worried about the quality of a basic device such as a smart light bulb. P14, who selected the no-name bulb, noted that the bulb is quite inexpensive as compared to a TV or refrigerator purchase. For the more expensive products, he considered the brand of the product, but for basic devices, for instance, smart lights, he goes for the cheapest option.

### 3.3.5. Post Task Questions

Once participants finished the main tasks, we asked them to answer seven questions and rate their knowledge before and after this user study on a 5-point scale (1: poor–5: very good).

Our results (Table 3) show that added information significantly improved the participants' general knowledge about smart devices from different aspects, such as technical and financial perspectives. The most notable increase in knowledge was reported in relation to paying for software and software/firmware warranties. Furthermore, all this additional information did cause some participants to change their original purchase decision, indicating that they would no longer purchase that smart device.

**Table 3.** Means of participant's knowledge before and after the user study.

| Question | Before | After | *p* |
|---|---|---|---|
| How do you rate your knowledge regarding the smart devices' privacy risk? | $M = 3.1\ SD = 0.78$ | $M = 4\ SD = 0.64$ | <0.001 |
| How do you rate your knowledge regarding the smart devices' performance risk? | $M = 3.2\ SD = 0.89$ | $M = 4.8\ SD = 0.79$ | <0.001 |
| How do you rate your knowledge regarding the smart devices' financial risk? | $M = 3.1\ SD = 1.02$ | $M = 4.15\ SD = 0.58$ | <0.001 |

**Table 3.** *Cont.*

| Question | Before | After | *p* |
|---|---|---|---|
| How do you rate your knowledge about the included sensors in the smart devices? | *M* = 3.05 *SD* = 0.94 | *M* = 4.2 *SD* = 0.83 | <0.001 |
| How do you rate your knowledge about the sensors' operation in smart devices? | *M* = 3.01 *SD* = 1.07 | *M* = 3.09 *SD* = 0.91 | <0.05 |
| How do you rate your knowledge about what you are paying not only for the hardware but also for the software? | *M* = 2.8 *SD* = 1.28 | *M* = 4.5 *SD* = 0.68 | <0.001 |
| How do you rate your knowledge about the software/firmware warranty on smart devices? | *M* = 2.65 *SD* = 1.09 | *M* = 4.2 *SD* = 0.77 | <0.001 |

## 4. User Study 2

While articles and books suggest anywhere from 5 to 20 participants as adequate for qualitative research [48,49], we decided to run the quantitative research with slight changes to the first user study as described below.

### 4.1. Methodology

The second user study was designed with a focus on financial risks to understand the customers' willingness to pay, that is, the maximum price a customer is willing to pay for a product. This study was piloted and carried out with 34 participants before distributing the questionnaire. The average completion time of the questionnaire was estimated through the pilot study. The questionnaire was distributed online using a survey platform, namely Amazon MTurk. This study used a convenient sampling method. We recruited 432 participants through the survey platform. A total of 432 full questionnaires were initially received. Based on an evaluation of completion time in the pilot study, questionnaires that had been completed in less than two minutes were excluded. We also removed questionnaires that selected mostly the same answer to the scaled measurement items. After applying the aforementioned criteria in the data screening process, 323 fully completed questionnaires were used for the analysis.

The participants were smart device users in the United States and Canada that had a good distribution of demographic characteristics, as shown in Table 4. A descriptive analysis suggested that 98.1% of participants believe themselves to be knowledgeable users of smart devices. Specifically, 76% of them have moderate and more knowledge about smart devices, and 47% of participants reported that they are very knowledgeable about smart devices.

**Table 4.** Second user study demographic characteristics.

| Demographic Charachteristics | Type | Frequency (n = 323) | Percentage (%) |
|---|---|---|---|
| Gender | Male | 157 | 48.6 |
| | Female | 162 | 50.2 |
| | Non-binary | 2 | 0.6 |
| | Prefer no to say | 2 | 0.6 |
| Age | Less than 30 | 56 | 17.3 |
| | 30–39 | 114 | 35.3 |
| | 40–49 | 70 | 21.7 |
| | 50–59 | 57 | 17.6 |
| | 60 and over | 26 | 8.0 |

**Table 4.** *Cont.*

| Demographic Charachteristics | Type | Frequency (n = 323) | Percentage (%) |
|---|---|---|---|
| Education Attainment | Less than high school | 2 | 0.6 |
| | High school graduate | 11 | 3.4 |
| | Some college | 38 | 11.8 |
| | Professional degree | 66 | 20.4 |
| | 2 year degree | 28 | 8.7 |
| | 4 year degree | 174 | 53.9 |
| | Doctorate | 4 | 1.2 |
| Income | Less than $10,000 | 1 | 0.3 |
| | $10,000–$19,999 | 28 | 8.7 |
| | $20,000–$29,999 | 33 | 10.2 |
| | $30,000–$39,999 | 37 | 11.5 |
| | $40,000–$49,999 | 46 | 14.2 |
| | $50,000–$59,999 | 49 | 15.2 |
| | $60,000–$69,999 | 24 | 7.4 |
| | $70,000–$79,999 | 27 | 8.4 |
| | $80,000–$89,999 | 18 | 5.6 |
| | $90,000–$99,999 | 16 | 5.0 |
| | $100,000–$149,999 | 32 | 9.9 |
| | More than $150,000 | 12 | 3.7 |
| Employment | Unemployed not looking for work | 9 | 2.8 |
| | Unemployed looking for work | 4 | 1.2 |
| | Student | 2 | 0.6 |
| | Employed part time | 33 | 10.2 |
| | Employed full time | 263 | 81.4 |
| | Retired | 12 | 3.7 |
| Smart Device Knowledge | Not knowledgeable at all | 6 | 1.9 |
| | Slightly knowledgeable | 72 | 22.3 |
| | Moderately knowledgeable | 93 | 28.8 |
| | Very knowledgeable | 113 | 35.0 |
| | Extremely knowledgeable | 39 | 12.1 |

In this experiment, we compared consumer responses through a 3 (warranty information: base, 5-year, lifetime) by 2 (smart devices: TV, fridge) between-subject user design. We predicted that consumers' willingness to pay (WTP) would be greater for a lifetime software update warranty (performance and financial risks). User WTP and devices' expected lifespan were the core dependent variable, and software update warranty was the independent variable. While in the first user study, the products were displayed with a price tag, and participants were aware that the price was within their budget to purchase, in the second user study, we asked participants to let us know their WTP for the product with the provided information, mostly with a focus on software update warranty. Financial

risks, performance risks, and privacy risks were tested as simultaneous mediators of the manipulations' effects on user preference.

### 4.2. Tasks

Each participant was randomly assigned to one of the following six conditions: TV/Fridge with no software warranty information (*TV/Fridge-Base*), TV/Fridge with a 5-year software update warranty (*TV/Fridge-5*), TV/Fridge with a lifetime software update warranty (*TV/Fridge-Life*). The questions are listed in Table 5.

**Table 5.** List of questions asked in the second user study.

| Question | Measuring |
| --- | --- |
| How much would you be willing to pay for this product?—Price ($) | WTP |
| Based on the available information on the image, how impressive is the warranty for this product? | Impressiveness |
| How long do you expect this product to last or flawlessly function under normal intensity of use (in years)? | Lifespan |
| Please indicate your opinions about the product that you are evaluating. (Attractiveness) | Attractiveness |
| Please indicate your opinions about the product that you are evaluating. (Recommend product to others) | Recommend product to others |
| Please indicate your opinions about the product that you are evaluating. (Purchase Intension) | Purchase Intension |
| How would you rate your overall knowledge of smart devices? | |
| How would you rate your overall knowledge that what you are paying for a smart product is not only for the hardware but also for software? | |
| How would you rate your overall knowledge of software updates on smart devices? | |

Once participants finished answering the questions, participants completed a demographic survey.

### 4.3. Results

The observed mean of WTP on TV-Life (*Mean*: 1051.11, *SD*: 320.9) was 12% higher than TV-5 (*Mean:* 941.84, *SD:* 285.63) (Figure 6). This difference is at the edge of statistical significance with an $\alpha$ level of 0.05 ($F(1,131) = 3.79$, $p = 0.053$). While the studied mean of WTP on TV-Life (*Mean:* 1051.11, *SD:* 320.9) was 9.9% higher than TV-Base, this difference did not reach statistical significance ($F(1,132) = 2.96$, $p = 0.08$).

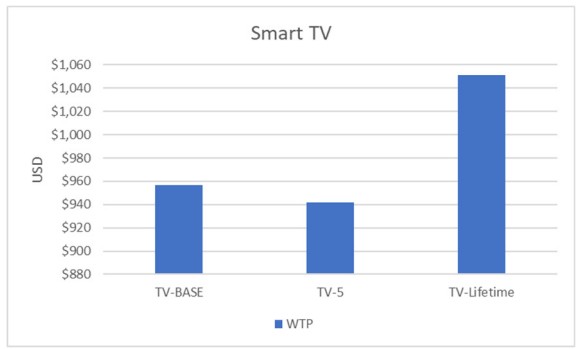
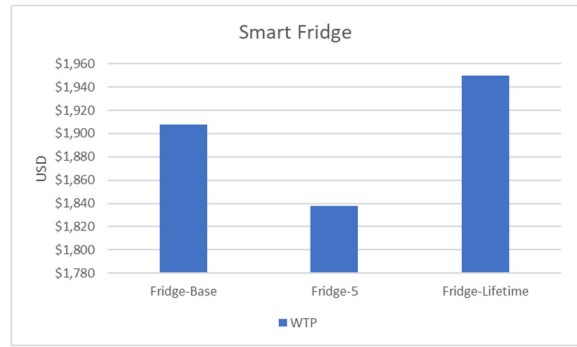

**Figure 6.** Willingness to Pay: Smart TV (**left**), Smart Fridge (**right**).

Whereas WTP was not significantly different from Fridge-Life (*Mean:* 1949.93, *SD:* 492.62) to Fridge-Base (*Mean:* 1907.85, *SD:* 435.91) and Fridge-5 (*Mean:* 1837.48, *SD:* 543.39), Fridge-Life's WTP displayed 2% and 6% increases compared with Fridge-Base and Fridge-5 (Figure 6).

## 5. Discussion and Future Work

Our initial analysis investigated how providing additional information will affect participants' decisions to purchase individual smart home devices. Based on their responses, we concluded that additional information about how monitoring sensors work and details about the software update warranty do affect consumer purchase behavior. Results from the second user study show that software update warranty information has an impact on the willingness to pay. As customers are ready to pay more for a lifetime warranty on a software update, we believe that the clarification between the product warranty and software update warranty could help customers to make a better decision and has an impact on WTP. To be fully transparent, instill trust, and address privacy risks, we believe that a list of sensors on smart devices needs to be available to consumers—just as similar energy efficiency labels are currently displayed on them. This will help customers understand more about the device that they are adopting and encourage them to ask about smart device sensors, if they are not listed, for future purchases.

While privacy risk is an important obstacle for buyers, transparency is key in the smart devices market, and customers will trust companies that do not hide sensor information about the devices that they sell. As such, our suggestion to IoT companies is to reveal as much as possible about the sensors they use in their devices, how the sensors work, how and what data will be collected, and where it will be stored.

In addition, we found that reference pricing for expensive devices is an important purchase decision parameter. Participants compared the price of a smart device with a non-smart device to find out how much they were paying for the smart features. Additionally, they will consider alternative options instead of built-in smart features. For instance, P13 mentioned that he could get a regular fridge for USD 2000 cheaper than a smart fridge and get an iPad with a proper holder installed on the fridge to have a smart screen on it. Other participants mentioned that if they found that the smart feature on a TV was more expensive and not reliable for a long period of time, they would go with a non-smart TV and use streaming dongles or set-top box solutions instead.

Almost all participants in the first user study identified that their older electronic smart devices are no longer working as fast as they used to, and they are equally worried about this problem for any smart home appliances they purchase. Expensive products and unfamiliar products, which are infrequent purchases such as a smart refrigerator or a robot vacuum, require high customer involvement, extensive time, and information research—which is a critical stage. During their research, customers are looking for full product information from the company or retailers that sell smart products so that they can make informed choices about which smart home products to buy.

We also found that additional features on higher-cost IoT products must be good enough for customers to justify paying more for a product with that feature. For instance, in a robot vacuum, which has only one application and is designed to do only one job (cleaning floors), automatic dirt disposal is a unique additional feature that will help customers to do less work so they will gain more value. Therefore, they are more likely to pay the additional cost for this feature.

On the other hand, customers' decision making for low-cost (and familiar) smart products needs less time, research, and thought (e.g., smart bulb purchases). As such, consumers are more likely to pay more for additional smart features (e.g., scheduling on/off for a smart bulb) as the cost of these added features is typically still well within their budget.

Because purchasing major appliances in a real-world scenario is complicated, customers may not find all the smart and non-smart features they are looking for in a single

product. Furthermore, unexpected features such as short-term software warranties that reduce the lifetime of the product make decisions harder. This effect is also identified in the second user study, while in both TV and Fridge, the WTP on a 5-year warranty was less than the base model. In addition, the lifetime warranty was liked by the participants, as this type of warranty extends the reliability of existing functionality and acts as a protection for the consumer against device failure. So, we suggest that the ideal solution for both manufacturers and consumers is to ease this decision-making process by guaranteeing that the device software will receive updates for the life of the product and that these updates will not reduce device performance over time.

Most of the participants in the first study identified as male. Due to the small sample size of female participants, we could not measure if there is a difference in smart home device purchase decisions of those who identify as female. We addressed this concern in the second user study, and with 48.6% Male and 50.2% female, we could not identify a significant difference in WTP and expected life span of the products between males and females. In addition, in all tasks, we set a price for each smart device and told participants to imagine that the price was within their budget. In reality, the socio-professional category and willingness to pay include the actual ability to pay and involve more power (and financial limitations) on purchase decisions. In future studies, we will explore these moderators.

Understanding the specific needs and considerations of both male and female IoT consumers, in addition to actual budget considerations, will provide even more valuable insights for smart home device Manufacturers, Designers, and Marketers to help increase the adoption of both their existing and new smart home products.

### 6. Conclusions

Our research shows that buyers are generally less aware of what sensors are used in smart devices, and (because the software of an IoT is not clearly visible to them) they do not typically realize that smart devices are made of two important integrated parts, hardware and software. For instance, in the second user study, 24% of participants reported that they have knowledge about smart devices, but at the same time, they have lower knowledge about software updates on those smart devices. Furthermore, they are largely unaware of software update requirements and that these updates can reduce the lifecycle and performance of their smart products over time. Being familiar with IoT devices in 2023 and understanding what factors need to be considered at the time of purchase could reduce the financial risks, performance risks, and privacy risks.

We also observed that providing additional information does indeed influence consumers' smart product purchase decisions. Specifically, how software update warranties and the type of sensors in smart devices' mediating performance point to financial and privacy risks, which can impact consumer purchase behavior. Other moderators, such as brand and product price, also have a direct effect on customer purchase decisions.

In summary, results from both qualitative and quantitative studies show that providing additional information about software updates and a list of sensors has an impact on consumer purchase preference. These results help smart device manufacturers and designers increase consumer trust in their brands. Addressing this knowledge gap could lead to a significant increase in consumer adoption of smart products and a huge increase in market potential for both existing and new IoT products.

**Author Contributions:** Conceptualization, M.N. and S.S.; methodology, M.N.; validation, M.N.; formal analysis, M.N.; investigation, M.N.; resources, M.N.; data curation, M.N.; writing—original draft preparation, M.N.; writing—review and editing, M.N. and S.S.; visualization, M.N.; supervision, S.S.; project administration, S.S.; funding acquisition, S.S. All authors have read and agreed to the published version of the manuscript.

**Funding:** This research was funded by Mitacs grant number Accelerate Grant And The APC was funded by Natural Sciences and Engineering Research Council.

**Data Availability Statement:** The data that support the findings of this study are available from the corresponding author, Srinivas Sampalli, upon request.

**Acknowledgments:** This research is funded in part by Mitacs and the National Sciences and Engineering Research Council of Canada.

**Conflicts of Interest:** The authors declare no conflict of interest.

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
