# Peer review of "Evaluating Consumer Behavior, Decision Making, Risks, and Challenges for Buying an IoT Product"

_2624-831X, doi:10.3390/iot4020005_

Round 1

Reviewer 1 Report (Previous Reviewer 1)

Thank you for this revised version of the paper. The additions and extensions of your study makes it -- at least -- honest in what the data can mean, and extend the validity of the data to my satisfaction.

Before I get to the details of the review, let me take this opportunity to dialogue a little with the authors on a general topic .. It appears that I disagree with what you call "a large number of articles and books" since the suggestion that a sample size of 5-20 can be representative of a population still seems incredibly reductionist to me: since we're in the space of "acquiring technology that is not cheap", it would seem that "disposable salary" and socio-professional categories would shape what people "know" about it.

For example, a college-educated adult in the tech-field would likely have both spent some time studying, and having had the disposable income permitting acquiring previous experiences with, the technology -- whereas a retiree with only a social security cheque likely would not.

Depending on which statistics institue you ask, they will suggest that there are around 30 distinct "socio-professional" categories -- which, would be crossed with 4-5 age-categories (from "child" through "college age", "young professional", "senior professional" and "retiree"). 

Drawing from 5 to 20 samples will mechanically not be able to span the gamut of both age-categories and SPCs -- and so, I maintain that your results from your first study are unlikely to be descriptive for the population at large.

It may be enough for a sub-population, alas, that sub-population has to be clearly called out -- and "our twitter followers" isn't quite good enough since it doesn't allow "reproducible research" (or follow-on research, to validate if things evolve, over a similar population).

And that's getting to the crux of my first comment, which is a follow-on from my previous review: your sample for the first study is drawn, and I understand how you did it -- I just don't understand how I could draw a sample with the same characteristics. You quantify gender and age -- that's good. But, as suggested, the "socio-professional category" of your test subjects is one I would expect be significant for the results. I do not see that (in section 3.1). I am dubious as to if "prior experience buying IoT products" is a suitable stand-in for that demographic factor. But that could be worth exploring (see below).

Your user study #2 corrects for this, by including more detailed demographic factors and I thank you for the effort of doing, and including it. However the introduction to study-2 is a little confusing. You start by saying that you have  done it over 34 participants (line 362) ... but then a little later (line 365) you talk about having recruited 432 participants, later reduced to (line 368) 323 "fully completed questionnaires" (I assume one per participant). Table 4 also lists an n=323

How did we get from 323 to 34? 

Next, what's the use of your study #2 in the paper? That's not actually a trick question. Section 4 presents (interesting) data, but sections 5 and 6 do not discuss them. Rather, I think that they a re content to discuss only the results of the 20-person study. Occasionally, you call that out explicitly (e.g. line 443, line 461) but otherwise it is not clearly stated what your discussions and conclusions are about. I strongly suspect that they are for only study #1 -- and that's a shame.

What I would have expected/hoped to see would be a discussion on what different aspects the two studies revealed? What you learned from doing the 2nd study? This could, for example, be that you got identical/concurring results with the 1st study, or that you were able to glean more details due to a different gender sampling, etc. It could be that "familiar with IoT devices" in 2023 is a good stand-in for specific socio-professional categories.

In short, you're selling yourself short by not exploiting and discussing the results of the considerable effort that study #2 would have been -- to the point that I find that you really need to do that before I can recommend the paper for publication.

With that in mind, I am recommending "Reconsider after major revision" -- with, to me, that major revision being that this discussion must be included for the paper to be publishable. However as it is a matter of analysing and discussing data that you have, I trust that you will be able to do so  -- and if you do, then that will allow me to recommend publication for your next version.

From my last review, I still have a couple of general writing-style comments to the paper.

Think about writing in a timeless fashion: the paper should be equally valid 10, 20, or 30 decades from now. At that point in time, what is “novel” today may have become part of “state of the art” (so, describing your own work as “novel” implicitly renders it obsolete as soon as the paper is published) — and, what is “recent” at the time of writing the paper, will be ancient history ... on the same token, what is “classic” at the time of writing may have become obsolete and irrelevant. What people have "started" to do today, is something that they'll have either stopped tomorrow, or "still be doing" by then. 

I do understand that any measurement of "a population's opinion on XXX" will be strongly time-dependent. And that's fine. You should make it clear that you did your study in 2022/2023, and it is perfectly fine to describe the state of the universe (that you're sampling) at that date. 

However that does not warrant the use of phrases such as "have started to add WiFi", or "increasingly wide variety" in such a general sense. That will -- most definitely -- have changed in 10 years.

What I am asking here is to sober up especially the abstract and the introduction in this regard. State that your study are about the decision making when acquiring IoT products. State that in 2022/2023, IoT products look/behave this way -- and that that's in contrast to how similar products looked/behaved previously.

That's factual, and anchored around the dates of your study -- and not around "Recently" (because, in 10 years, your study won't be recent...).

I'd also encourage -- though this is hard, I recognize -- that you try to consider what lasting learnings can come from your study. Will the conclusions about human behaviours on "acquiring IoT devices" also be applicable when in 10 years time the question is "acquiring a flying car"?

While I recognise improvements on this next issue, I still find that you have a lot of "assertions and claims": "common IoT devices". "preloaded to the microcontroller of any IoT device", "consumer knowledge is surprisingly limited". I'd strongly suggest that you review the paper yourselves and every time you "make a claim" try to find a way of justifying it -- and if you can't, remove the claim.

Finally, there're a couple of phrases here and there that read like you were writing them while coffee-deprived -- likely because you had just been analysing >300 questionnaires from your test subjects ;) So I'd suggest that a review for clarity and simplicity (especially of the last paragraph of the abstract, where you talk about your two studies) would be a good idea.

Author Response

Thank you for your very valuable comments. Please see our detailed response to each comment in the attached file.

Reviewer 2 Report (Previous Reviewer 2)

Section 4 introduces the new study with a larger sample, considering my comments in the previous version.

However, sections 5 and 6 did not change and considered only the first study, which is unacceptable. You must change those sections to include the experiment in section 4.

Author Response

Thank you for your valuable comment. Please see our response in the attached file.

Round 2

Reviewer 1 Report (Previous Reviewer 1)

Thank you for this revision to your paper, and for having addressed my concerns - particularly those related to your study #2, which is now much much clearer.

I have a few remaining nits. Unfortunately you did not include line numbers, so I will try to present them textually, (where possible) bold and underlined the words that I recommend be substituted:

"As in 2022/2023, smart technology is still relatively new compared to traditional unconnected..." -> "In 2022/2023...."

"While articles and books suggest anywhere from 5 to 20 participants as adequate for qualitative research [14,24], we decided to run quantitative research to with slightly changes to the first user study as described below." -> "slight changes"

"Task’s steps" in several headlines -- generally, I recommend against phrasings in possessive form, as it becomes slightly harder to read. Why not just "Steps"? -- likewise, I actually would prefer to not have this be a sub-sub-sub-sub-section, each time: you already have bullets, so simply a lead-in sentence" Ala "this part of the survey had the following steps"?

Author Response

Thank you for your kind suggestions. We have addressed all your comments. Please see the attached file for the response to each comment.

Reviewer 2 Report (Previous Reviewer 2)

In this revised version there is now some discussion about the second experiment, and it is also mentioned in the conclusions.

However, the overall text is not uniform, the second experiment is more significant than the first but is less explained and less discussed. It would be best if you discussed each experiment separately, and then you may compare them in terms of the different or similar information they give you.

The paper is being patched in each version, but it must also be reorganized, otherwise, it is not publishable.

Author Response

Thank you for your valuable comments and suggestions. We have addressed all your suggestions. Please find attached a file that provides the details of our responses.

This manuscript is a resubmission of an earlier submission. The following is a list of the peer review reports and author responses from that submission.

Round 1

Reviewer 1 Report

This paper is not at all technical -- rather, it is a consumer product survey. It seems to perhaps fit within the "IoT User experience" and the "IoT Business aspects" categories of the journal.

That aside, I do question if the results actually are "deep enough" to merit publication. The questions that you ask are interesting, but I am not convinced that the experiments that you conduct are exhaustive enough -- basically, you are sampling over a population of just 20 participants -- to provide significant conclusions. Further, it appears that your sampling might be biased to a specific population segment, namely  university students & those that follow you on social media (presumably overwhelmingly students and academic). In order to recommend the paper for publication, I would want to see either a more thorough analysis of this bias, and the consequences thereof -- or a less biased sampling method.

I have a couple of general writing-style comments to the paper

Think about writing in a timeless fashion: the paper should be equally valid 10, 20, or 30 decades from now. At that point in time, what is “novel” today may have become part of “state of the art” (so, describing your own work as “novel” implicitly renders it obsolete as soon as the paper is published) — and, what is “recent” at the time of writing the paper, will be ancient history ... on the same token, what is “classic” at the time of writing may have become obsolete and irrelevant. What people have "started" to do today, is something that they'll have either stopped tomorrow, or "still be doing" by then.

Get rid of “value-ridden words” or phrases, such as “often very much higher than XXX”, “simply doing XXX”, etc.  If such statement is true, and it may well be, then provide a quantitative argument here, or a citation, or both (preferable). So when you say "Many home appliances" or "they often don't realise", etc, please quantify.

Generally, when using "like XXXX", consider using "such as XXXX" instead -- it's less "spoken language"

You have a tendency to "write in conditionals" that actually obscures what you're trying to convey -- I think. For example:

"it is expected that customers would not be completely aware" -- what are you trying to convey by "would not be"? Are they, or are they not expected to be aware? Why not simply "it is expected that customers are not completely aware"? Same thing with "could/should/would/*", they leave ambiguity.

The introduction is well written, but there're quite a few assertions presented, where you claim something without actually bringing proof. for example:

"To illustrate, when they first came out, many of these smart home VA devices were promoted as “Works with Alexa” or “Works with the Google assistant” labels. Now the wording has changed to, “Alexa built-in” or “Google Assistant built-in”."

Where's the data affirming this to be fact -- rather than something you pull out of a hat? (And also, this is -- if anything -- not timeless, but rather assume a very specific snapshot in time ; see above). This is but one example, please tighten up the paper on this point. 

Another worse example of this is in line 81, actually, since you pretty much is using an unsubstantiated assertion as the basis for your paper:

"Research shows that the consumer decision-making process involves five steps: 1) ... 2) .... 3) ... 4) ... and 5). Our research focuses on steps 2 to 4.".

Please-oh-please provide citations to the research that shows that -- otherwise, you're basing your work on nothing.

The whole section starting in line 130 is, likewise, nothing but an unsubstantiated assertion: no citations given for a massive amount of postulates (that may or may not be true)

I am confused by your citation style. Sometimes you cite just [42] -- whereas other times you do "Dupont et.al show that ... [42]". What gives? Pick the former, and stick to it.

I'd also like to encourage that you get rid of “value-ridden words” or phrases, such as “often very much higher than XXX”, “simply doing XXX”, etc.  If such statement is true, and it may well be, then provide a quantitative argument here, or a citation, or both (preferable). For example, you talk about "much research has been done to evaluate..." - how do you quantify "much"? ;) And where do you get that evaluation? This is just one example, again, please tighten up your arguments by avoiding this.

Details

- I am uneasy by you yourself declaring your work for "valuable" (line 97). It comes across as "marketing"

- Line 173, mean net personal annual income does not mean much -- or rather, it is extremely context-dependent. In a rural backwater, 50KUSD may be a significant salary -- however in the California Bay Area, 50KUSD would be well below the poverty line.

- Line 178, is the use of Zoom significant to your findings? If not, then just say "using videoconferencing"...you're not the Zoom marketing department, are you? ;) Idem for line 181, is the fact that it was a Google form significant vs a Limesurvey or whatnot? If not, then just say "an online form", unless you are also the Google marketing department. This is not independent from the "Timeless" comment previously -- it is plausible that in the future, neither of these two companies will exist or be known, and your paper should remain valid even then.

- Line 207, line 223, and for the remainder of the paper there's a large number of weird "Error! Reference source not found" messages appearing.

- Line 208 "VGA camera" -- really? First , is the resolution of the camera of significance to your work? And if not, then why mention it. Second, if it indeed is VGA cameras that are on these robot vacuums, then I'd be surprised, that resolution has been passe for more than 2 decades by now.

- It seems that the Funding. data availability, and acknowledgements sections needs a revision

Author Response

Please find the authors' responses to the reviewers’ comments. 

Reviewer 2 Report

The sample size is too small, you may consider it as preliminary results to guide future research, but it is not enough for a journal article. It is also not representative, with only up to two women in the study.

However, it is an interesting approach, and with further research and a more representative and larger sample, I encourage you to submit another article.

Some suggestions and text errors:

- line 15: affect, not effect.

- line 135: "of" before "software".

- line 200: should refer to the light bulb in this part of the text.

- several lines: lots of broken references (Error! Reference source not found.).

- line 294: move to the next page.

- line 368: "may might" -> might.

- line 377: believes (s missing).

Author Response

(The authors gave the same response as above.)
